# Learning active tactile perception through belief-space control

Jean-François Tremblay, Johanna Hansen, David Meger, Francois Hogan, Gregory Dudek

*Abstract*— **Robots operating in an open world can encounter novel objects with unknown physical properties, such as mass, friction, or size. It is desirable to be able to sense those property through contact-rich interaction, before performing downstream tasks with the objects. We propose a method for autonomously learning active tactile perception policies, by learning a generative world model leveraging a differentiable bayesian filtering algorithm, and designing an information-gathering model predictive controller. We test the method on two simulated tasks: mass estimation and height estimation. Our method is able to discover policies which gather information about the desired property in an intuitive manner.**

## I. INTRODUCTION

Robots operating in an open world can encounter arbitrary, unseen objects and are expected to manipulate them effectively. To achieve this, robots must have the ability to infer the physical properties of unknown objects through physical interactions. The online measurement of these properties is key for robots to operate robustly in the real-world with open-ended object categories.

Psychology literature refers to the way humans measure these properties as *exploratory procedures* [1]. These procedures, for example, include pressing to test for object hardness and lifting to estimate object mass. These exploratory procedures are challenging to hand-engineer and vary based on the object class. This work focuses on learning such exploratory procedures to estimate object properties through belief-space control. Using a combination of 1) learning-based state-estimation to infer the property from a sequence of observations and actions 2) information-gathering model-predictive control (MPC), we demonstrate that it is possible to learn to execute actions that are informative about the property of interest and to discover exploratory procedure without any human priors.

## II. RELATED WORKS

### A. Learning for state-estimation

There are several works proposing the fusion of Bayesian filtering methods with deep learning, where the dynamics and observation models used are learned neural networks.

Lee et al. [2] provide a good overview of learning Bayesian filtering models for robotics applications, and release `torchfilter`, a library of algorithm for this purpose which we build on for our belief-space control algorithm.

In [3], the authors present the Backprop Kalman filter described as a discriminative approach to filtering. Discriminative filtering does away with learning an observation model (a mapping from state to observation) and learns a mapping from observation to state instead. Here, we argue that learning a generative observation model, while more

computationally challenging, is key to predicting future state uncertainty and planning for informative actions.

Burkhart et al. [4] present the discriminative Kalman filter concurrently to [3]. This approach assumes linear dynamics and models the prior over observations as Gaussian. It can only handle stationary observation processes.

### B. Active perception

Active perception consist of acting in a way that assists perception and can incorporate learning, including the learning methods above. Denil et al. [5] use reinforcement learning for "Which is Heavier" and "Tower" environments, where the goal of the former is to push blocks and, after a certain interaction period, take a "labelling action" to guess which block is heavier. You get a reward if the label is correct. They then train a recurrent deep reinforcement learning policy on that environment. The action space for these problems is constrained and designed to act such that the blocks are pushed with a fixed force towards their center of mass. While this method enables the robot to effectively retrieve mass using human priors and intuition, our work differs where the robot is tasked with discovering such behaviors autonomously with unconstrained action spaces.

More specifically to robotics, Wang et al. [6] introduce SwingBot, a robotic system that swings up an object with changing physical properties (moments, center of mass). Before the swing up phase, the system follows a hand-engineered exploratory procedure that shakes and tilts the object in the hand to extract the necessary information for a successful swing up. Rather than engineering the exploration phase, we propose a generic framework for extracting such information before accomplishing a given task.

## III. METHODS

We are in a controlled hidden Markov model (HMM) setting (a partially observable Markov decision process (POMDP) without a reward function), where each observation $o_t$ gives us partial information about the state of the robot and object we are interested in. More formally a controlled HMM is a tuple $(\mathcal{S}, \mathcal{A}, p(s_{t+1}|s_t, a_t), \Omega, p(o_t|s_t))$, where the state, action and observation space ($\mathcal{S}$, $\mathcal{A}$ and $\Omega$ respectively) are in $\mathbb{R}^n, \mathbb{R}^m, \mathbb{R}^d$ respectively. It is important to note that in this context, the state can contain robot pose and velocity, object pose and velocity, object properties, and all properties that describes the environment and that are subject to change either during or in between episodes. The representation for the state will be learned in a self-supervised fashion, as described in § III-A, and will be

learned in such a way that the first element of the state represents the object property of interest:

$$s_t = (m_t, z_t), m_t \in \mathbb{R}, z_t \in \mathbb{R}^{n-1}. \tag{1}$$

We are in an episodic setting with ending timestep $T$, and where at each episode the object is randomized. For mass estimation as an example, at each episode, an object with a different mass is presented and the goal is to infer the mass of this new object.

In § III-A we describe how to infer the belief state (containing an estimate of the object property of interest) $b_t \approx p(s_t|a_0, \ldots, a_{t-1}, o_1, \ldots o_t)$, $\bar{b}_t \approx p(s_t|a_0, \ldots, a_{t-1}, o_1, \ldots o_{t-1})$. In § III-B we use that estimate to design an information-gathering controller. Finally, in § III-C we present how to integrate these two things in a data-collection/training and control loop.

### A. Learning-based Kalman filter

Here the goal is to learn a dynamics and observation model while performing belief-state inference. The dynamics model representing $p(s_t|s_{t-1}, a_{t-1})$ is

$$s_t = f_\theta(s_{t-1}, a_{t-1}) + \Sigma_\theta(s_{t-1}, a_{t-1})w_t \tag{2}$$

where $w_t$ are independent and identically distributed (IID) standard Gaussian random variable in $\mathbb{R}^n$.

Generative filtering (as opposed to discriminative filtering [2, 3]) implies learning a generative world-model, able to fully simulate the system and generate observations via the equation

$$o_t = h_\theta(s_t) + \Gamma_\theta(s_t)v_t. \tag{3}$$

where $v_t$ are IID standard Gaussian random variables in $\mathbb{R}^d$. While learning this model can be more challenging in the face of high-dimensional and complex observation spaces (e.g. images), it opens up new avenues for forward belief-space planning.

Using an explicit-likelihood (Gaussian state-space model) setting, we train the model in an self-predictive manner. In (8), we present the derivation for the loss of the generative observation model. This derivation is adapted from [7] Chapter 12, where we integrate action variables.

$$p(o_1, \ldots, o_T|\theta, a_0, \ldots, a_{T-1}) \tag{4}$$

$$= \prod_{t=1}^T p(o_t|\theta, o_1, \ldots, o_{t-1}, a_0, \ldots, a_{t-1}) \tag{5}$$

$$= \prod_{t=1}^T \int_{\mathbb{R}^n} p(o_t|\theta, s_t)p(s_t|\theta, o_1, \ldots, o_{t-1}, a_0, \ldots, a_{t-1})ds_t \tag{6}$$

$$\approx \prod_{t=1}^T \int_{\mathbb{R}^n} p(o_t|\theta, s_t)\bar{b}_t(s_t|\theta)ds_t \tag{7}$$

$$= \prod_{t=1}^T \mathbf{E}_{s_t \sim \bar{b}_t(s_t|\theta)} p(o_t|\theta, s_t) \tag{8}$$

Here $\bar{b}_t$ is the output of the predict step of our filter with input $b_{t-1}$ and $a_{t-1}$. It is only an approximation of

$p(s_t|\theta, o_1, \ldots o_{t-1}, a_0, \ldots, a_{t-1})$. If we take the log, get a lower bound from Jensen's inequality and compute the empirical mean, we get:

$$\log p(o_1, \ldots, o_T|\theta, a_0, \ldots, a_{T-1}) \tag{9}$$

$$\gtrapprox \sum_{t=1}^T \frac{1}{N} \sum_{i=1}^N \log p(o_t|\theta, s_t^i) \quad s_t^i \sim \bar{b}_t(s_t|\theta) \tag{10}$$

$$:= \text{ELBO} \tag{11}$$

Equation 9 gives us a lower bound of the log likelihood (similarly to the ELBO loss in VAEs [8]) to train our model leveraging the differentiable approximate inference used to compute $\bar{b}_t$. Because $\bar{b}_t = \mathcal{N}(s_t|\bar{\mu}_t, \bar{\Sigma}_t)$, we can use the reparametrization trick to sample $s_t^i$ by sampling $\xi^i$ from a $n$-dimensional standard Gaussian, and then letting

$$s_t^i = \bar{\mu}_t + \bar{\Sigma}_t \xi^i \tag{12}$$

$\theta$ represents the parameters for $f, \Sigma, h, \Gamma$ which are neural networks. We jointly perform state-estimation and parameter optimization by estimating $b_t = (\mu_t, \Sigma_t)$ using a extended Kalman filter (EKF), the operation of which are all differentiable (as shown for example by Lee et al. [2]), and maximizing the likelihood of the ground-truth object property of interest. For example, if mass is of interest, the loss for one timestep for an episode where the ground-truth mass is $m$ would be:

$$\mathcal{L}_m = -\sum_{t=1}^T \log \mathcal{N}(m|\mu_t^1, \Sigma_t^{11}) \tag{13}$$

Where $\mathcal{N}(\cdot|\mu, \sigma)$ is a Gaussian pdf with mean $\mu$ and variance $\sigma$. The first element of the state represents the mass, and we are maximizing its log-likelihood.

The loss we minimize is a combination of the self-predictive loss for the observation, and the likelihood of the mass in the state representation:

$$\mathcal{L} = \text{ELBO} + \mathcal{L}_m \tag{14}$$

In practice, we sample sequences of length less that $T$, and initialize the filter using stored beliefs in the dataset, in a truncated backpropagation through time fashion.

### B. Information-gathering model-predictive controller

The goal is to control the belief space process in a way that collects information about the property we're trying to perceive. The belief space for continuous systems is generally infinite dimensional (the space of probability distributions over the state space) thus intractable to work with using traditional control tools. However, by approximating the belief space using a parametric family (a Gaussian in our case), the problem can be formulated as a standard finite-dimensional continuous control problem. This is what we tackle here.

*a) Belief dynamics:* We can use the learned world model to simulate the belief space dynamics, as illustrated in Figure 1. The key is to be able to use the learned observation model to predict the future uncertainty about the state, rather than merely predict future states.

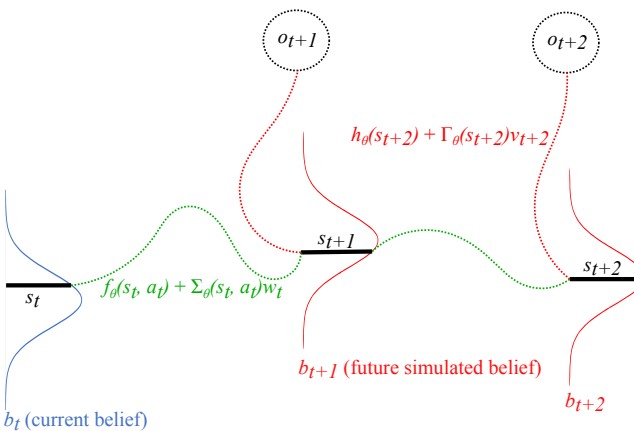

Fig. 1. Illustration of the sampling process for belief-space planning using a generative model. First, states are sampled from the current belief. We can then use our dynamics model and candidate actions to sample future states. These future states are given to our generative observation model to generate observations. We can then feed the generated observations and candidate actions to the state estimator to simulate the belief-space dynamics.

*b) Cost function:* We want our controller to minimize the entropy $H$ of the system:

$$J = \sum_{t=1}^{T} H(b_t^1) \qquad (15)$$

to minimize the uncertainty about the property of the object as soon as possible in the episode (compared to a final cost formulation). Minimizing this cost, for a Gaussian belief $b_t = (\mu_t, \Sigma_t)$, is equivalent to minimizing the cost

$$J = \sum_{t=1}^{T} \log \Sigma_t^{11} \qquad (16)$$

*c) Optimizer:* In this work, we used a sampling-based optimizer which selected the randomly-generated sequence of actions, minimizing the cost. The actions were generated using a Gaussian random walk in three dimensions, with a standard deviation of 10 cm. Following the model-predictive control framework, we only execute the first action of the sequence and then re-optimize.

### C. Full training and control loop

During training, we follow the procedure:

1) Collect data using current controller for one epoch (randomizing the object property of interest), saving the observations, actions and estimated beliefs as well as the ground truth object property for this epoch
2) Train the state estimator using the dataset
3) Update stored beliefs in the dataset (by replaying the actions and observations)

Step 3) does not have to be done every epoch and can be costly as the dataset grows, but it is important to perform truncated backpropagation through time and initialize our state estimate during training.

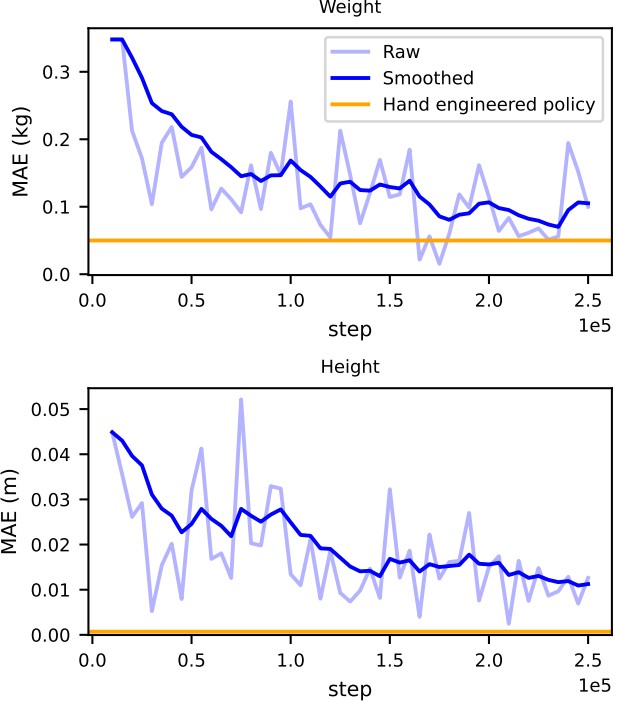

Fig. 2. MAE for the property estimation tasks, at the end of the episode averaged over 5 runs, as learning progresses. The hand engineered policy gives an upper bound on what can be achieved when the behavior must not be discovered, and we simply have to extract the mass from a sequence of sensor readings.

## IV. EXPERIMENTS

We set up a custom robosuite [9] environment for our experiments. The robot is a Franka Emika arm with a palm-shaped end-effector (as shown in Figure 3) and a force-torque sensor at the wrist. At each episode, a cube of the same size and visual appearance is laid down at the same location, with only its mass changing. We use position control, only translation. The observations are low-level for now: joint pose and velocity, object pose, force and torque at the wrist.

### A. Mass estimation

The first task is to learn to estimate the mass of a cube. The cube has constant size and friction coefficient, but its mass changes randomly between 1 kg and 2 kg in between episodes. Because the robot has no gripper, just a palm, it can't pick up the object, but it should be able to push it and extract mass from the force and torque readings generated by the push.

### B. Height estimation

The second task is to learn to estimate the height of a block, randomized between 1 cm and 15 cm. The force torque sensor, in this scenario, also acts has a contact detector. The expected behavior would be to come down until contact is made, at which point you can extract the height from forward kinematics (keep in mind that our

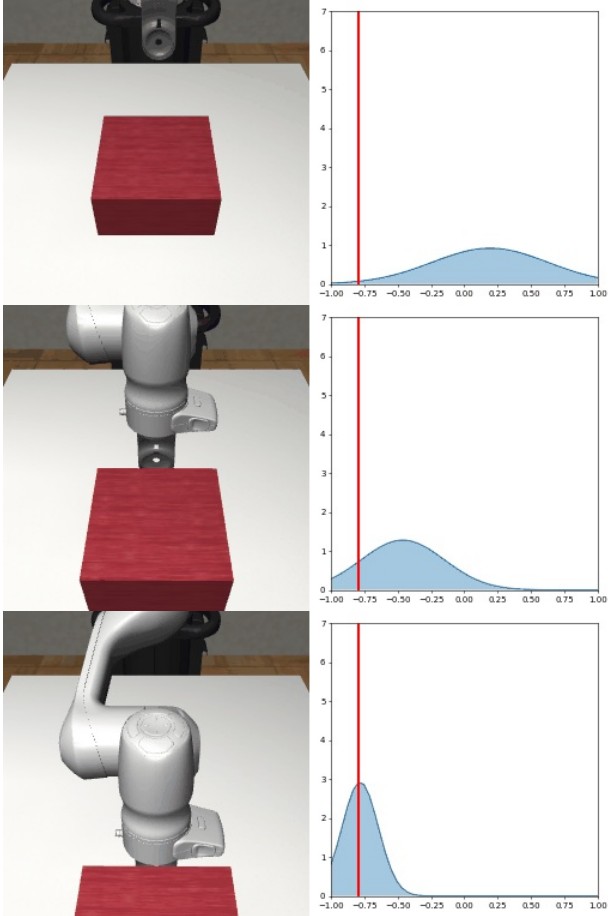

Fig. 3. Demonstration of the learned controller for mass estimation. We can see that it learns to stably push the object to extract mass from force torque readings. Notice how the uncertainty goes down as the arm starts pushing the block.

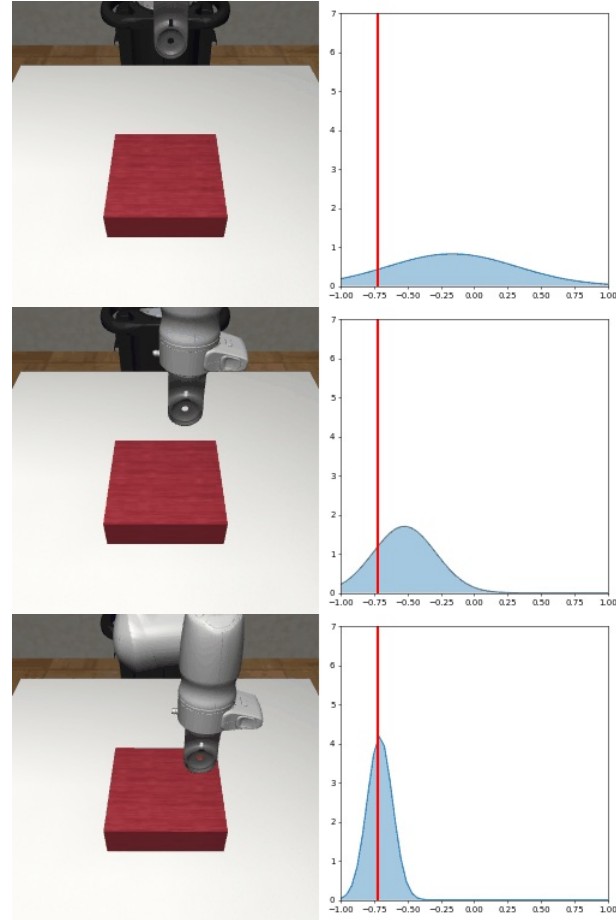

Fig. 4. Demonstration of the learned controller for height estimation. We can see that it learns to come down and adjust its estimate as it moves through free space, until touching the block.

method has no concept of forward kinematics embedded into it). One subtlety is that the arm must position itself above the box before moving down, as it can otherwise make contact with the table instead.

## V. RESULTS

Every 5000 environment steps, we run the evaluation procedure. It consists of running 5 episodes with randomized object property, and computing the MAE, where the absolute error is computed using the estimate at the last timestep of the episode. The training curve, showing the evolution of the MAE for the different tasks is shown in Figure 2. In the graph, a line is shown where a information-gathering policy was hand-coded by a human and we trained the state-estimator; straight pushing for mass and coming down to touch the block for height. It is meant as an approximate upper-bound for the information-gathering controller.

We can see that as learning progresses, two things happen concurrently:

1) the agent learn to perform informative actions. In the case of mass estimation, the policy pushes the block stably as shown in Figure 3. In the case of height

estimation, the policy goes down in a straight line until it touches the blocks.
2) the state-estimator learns to extract mass from the raw observations generated by the informative actions. For example during height estimation, the uncertainty remains high until the end-effector touches the block, at which point the estimate peaks at the correct height.

It is important to note that the pushing strategy is in no way encoded in the agent; initial trajectories are simply random walks in the workspace.

## VI. CONCLUSION

With the goal of discovering active tactile perception behaviors to measure object properties, we designed a learning-based state estimator and an information-gathering controller. Together, these two pieces allowed a simulated robot to discover a pushing strategy for mass estimation and a top-down patting strategy for height estimation, without any prior on what should the trajectory be. This opens up the door to learning more complex information-gathering policies, such as those for estimating the center of mass, hardness, friction coefficient and more.

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
