# OpenReview forum: "Learning active tactile perception through belief-space control"
_ICRA.org/2022/Workshop/Contact-Rich — ICRA 2022 Workshop: RL for Manipulation Poster_

### Official Review · Reviewer_PomW · 2022-05-05

**Rating:** 6
**Confidence:** 4

**Review:**

### Summary
This paper presents a method to learn tactile-based policies. The proposed method includes learning a generative world model and designing an information-gathering model predictive controller.
The main idea is to learn to identify the physical properties of interest of an object by interacting with it. In other words, to learn to estimate some physical property of an object of interest by simultaneously learning information-gathering actions.

The topic is interesting and relevant to the workshop.

### Comments

- The contribution of this work is not very clear. Consider giving a more explicit explanation to what is the concrete contribution(s) of the paper. Is it just the combination of learning-based state-estimation and information-gathering model predictive control? perhaps there is more but it is just not clear enough.
- Explanation of the method in Section III.A is not very clear. Consider more explicitly explaining each component of each new equation or at least citing the source of the formulations.
- Figures are not very clear. In Figure 2.: the abbreviation MAE is never introduced, in the entire paper. In Figures 3. and 4. the plots have no legends and the x-axis goes from -1 to 1, what do they mean exactly? Also a suggestion for the images in the figures, a side-view would be more informative to the reader. The front view does not clearly show the progression of the robot.
- For the experimental part, a baseline would highlight the achievements of your method. Additionally, though the target tasks are interesting, a better discussion of the results is necessary. Currently, section V. is very short and vaguely describes the results.
- Consider discussing the strengths and limitations of your method.

---

### Official Review · Reviewer_pRmM · 2022-05-12
**Learning active tactile perception through belief space control - review**

**Rating:** 7
**Confidence:** 4

**Review:**

This paper develops a controller in parallel with a learning-based Kalman filter to form estimates of unknown object properties such as mass and height that can be learned through interaction. They include a derivation of the loss function used to train a learned dynamics model that simulates belief estimates of future states, and design a controller that aims to minimize the entropy of the system (i.e. the uncertainty of the parameter estimate). The mass and height estimator and controller are both evaluated on a robosuite task which is just able to interact with an object via pushing and extract force/torque readings. In their experiments, they show that their method indeed extracts the relevant height and mass information by learning to perform exploratory actions that update the information of the estimate. It is unclear what initial policies were chosen for both of these tasks, as the hand-coded information-gathering controller would have likely contained the necessary exploratory actions needed for the controller to converge to a policy that extracts the desired input. However, overall this paper explores an interesting initial idea, and would be interesting to see expanded into a more extensive future work that estimates more than just mass and height.